# Biosorption of Neodymium (Nd) from Aqueous Solutions Using *Spirulina platensis* sp. Strains

**DOI:** 10.3390/polym14214585

**Published:** 2022-10-28

**Authors:** Éder C. Lima, Diana Pinto, Matias Schadeck Netto, Glaydson S. Dos Reis, Luis F. O. Silva, Guilherme L. Dotto

**Affiliations:** 1Institute of Chemistry, Federal University of Rio Grande do Sul, Porto Alegre 90040-060, Brazil; 2Department of Civil and Environmental, Universidad De La Costa, Calle 58 # 55-66, Barranquilla 080002, Colombia; 3Research Group on Adsorptive and Catalytic Process Engineering (ENGEPAC), Federal University of Santa Maria, Santa Maria 97105-900, Brazil; 4Department of Forest Biomaterials and Technology, Biomass Technology Centre, Swedish University of Agricultural Sciences, SE-901 83 Umeå, Sweden

**Keywords:** biomass, *Spirulina*, rare earth elements, kinetics, isotherms

## Abstract

Rare earth elements such as neodymium (Nd) are important elements used mainly in developing new technologies. Although they are found in low concentrations in nature, they can be obtained by extracting solid samples such as phosphogypsum. Among the techniques, adsorption has been used successfully with several adsorbent materials. In this work, two strains of *Spirulina platensis* (LEB-18 and LEB-52) were employed as biosorbents for efficiently removing the Nd element from the aqueous media. Biosorption tests were carried out in a batch system, and the results of the biosorption kinetics showed that for both materials, the biosorption of Nd was better described by the Avrami model. Moreover, it could be considered that 80 min would be necessary to attain the equilibrium of Nd(III) using both biosorbents. The result of the biosorption isotherms showed that for both strains, the best-fitted model was the Liu model, having a maximum biosorption capacity of 72.5 mg g^−1^ for LEB-18 and 48.2 mg g^−1^ for LEB-52 at a temperature of 298 K. Thermodynamics of adsorption showed that for both LEB-18 and LEB-52 the process was favorable (∆G° < 0) and exothermic (∆H° −23.2 for LEB-18 and ∆H° −19.9 for LEB-52). Finally, both strains were suitable to uptake Nd, and the better result of LEB-18 could be attributed to the high amount of P and S groups in this biomass. Based on the results, a mechanism of electrostatic attraction of Nd^3+^ and phosphate and sulfate groups of both strains of *Spirulina platensis* was proposed.

## 1. Introduction

Rare earth elements (REEs) are 17 elements, which include neodymium (Nd). Nowadays, REEs have been important for developing new technologies such as LED monitors, mobile phones, lasers for medical applications, LED bulbs, metallic implants, and other uses [1,2,3]. In addition, REEs are important because of their catalytic, magnetic, and phosphorescence properties [1,4]. However, one of the problems of REEs is their low content in nature, whose values are on the scale of µg g^−1^ [1]. Conversely, phosphogypsum is a waste from the production of phosphoric acid, which is rich in REEs [5,6]. Therefore, founding a strategy to extract REEs from a secondary source becomes very important [5,6,7,8].

There are several methods for the extraction of REEs from solid samples, such as bioleaching [9], electrochemical extraction [10], solvent extraction [11], and sulfuric acid and fluoride solutions as extractants [12]. However, the REEs must be recovered from the liquid and concentrated after extraction from the solid to the liquid phase. In this scenario, adsorption is a suitable way [13,14,15,16,17]. This separation method is one of the most important for extracting species from liquid media because the retained specie is concentrated in a solid phase. Afterward, a volume of an extract is passed by the solid phase, removing the retained specie to a volume that is considerably lower when compared to the initial volume [18,19]. Thus, several adsorbents have been utilized for the uptake of REEs, such as zeolites [20], amino-functionalized silica materials [21], composite materials of CaCO_3_:Eu^3+^@SiO_2_ [22], organofunctionalized minerals [23], polymers [24], algae [25], fungus [26], and others.

*Spirulina platensis* blue-green microalga can be produced in large amounts worldwide and is relatively inexpensive for biosorption [27,28,29,30]. This biomass is an alternative source of protein for human food and feed purposes [27,28,29,30]. It is rich in lipids, polysaccharides, and vitamins [30]. *Spirulina platensis* has different functional groups, such as hydroxyl, carboxyl, phosphate, and sulfate [30], that can bind REEs. *Spirulina platensis* has been used as a biosorbent for removing organic molecules [27,28,29,30] and heavy metals [31,32,33]. Although REEs have been uptaken utilizing biosorbents, up to the best of our knowledge, this paper is the first to report the Nd uptake using *Spirulina platensis* (blue-green microalgae).

In this paper, two strains of *Spirulina platensis* (LEB-18 and LEB-52) were employed as biosorbents for the removal of Nd elements from aqueous effluent. The differences in the number of functional groups between the two strains have a role in removing Nd^3+^ from aqueous solutions.

## 2. Materials and Methods

### 2.1. Biomass Preparation and Characterization

As previously described, *S. platensis* strain LEB-52 was cultivated under uncontrolled conditions in 450 L open outdoor photo-bioreactors [28]. During these cultivations, water was supplemented with 20% Zarrouk synthetic. As a result, an initial biomass concentration was 0.15 g L^−1^. In parallel, the *S. platensis* strain LEB-18 culture was inoculated in a 1 L photo-bioreactor with an initial *Spirulina* sp. concentration of 0.15 g L^−1^ [27]. Strain LEB-18 was cultivated in 20% Zarrouk medium and diluted with sterilized Mangueira Lagoon water [27]. Finally, both biomasses were dried in a tray and grounded to obtain particle size lower than 125 μm.

The biomasses were characterized according to the centesimal chemical composition [27,28]. The zero point charge (pHzpc) of *S. platensis* was determined [34]. Fourier transform infrared spectroscopy technique was used to examine the surface functional groups of *S. platensis* (Prestige 21, the 210045, Japan) [35]. SEM images were acquired on a Vega 3XM Tescan microscope. The elemental composition of the biomasses (C, N, H, O, P, S) was carried out in an Elemental Analyzer (brand Elementar, model Vario Macro Cube). The surface area (S_BET_) and total pore volume (V_TOT_) were measured using a Micromeritics 3Flex physisorption instrument (Micromeritics Instruments, Norcross, GA, USA).

### 2.2. Biosorption Assays

Neodymium nitrate hexahydrate (Nd(NO₃)₃.6H₂O; 99.9% purity) from Sigma Aldrich was used to prepare Nd(III) stock solution of 1.00 g L^−1^, which was diluted to desired concentrations to perform the experiments. Adsorption experiments on the LEB-18 and LEB-52 *Spirulina platensis* algae were conducted in batch mode using Erlenmeyers and a thermostated agitator (Marconi, Piracicaba, Brazil). The fixed conditions were: an adsorbent dosage of 2.00 g L^−1^, a volume of the solution of 50 mL, and a stirring rate of 200 rpm.

At first, the initial pH effect on Nd(III) adsorption was evaluated from 1.0 to 6.0. In these experiments, each microalgae material was inserted in Nd(III) solutions (20 mg L^−1^) and stirred for 2 h at 298 K. Kinetic tests were carried out at pH 6.0, varying the contact time from 0 to 240 min. Isotherms were carried out at a pH of 6.0 but with stirring of 24 h, from 298 to 328 K, and an initial concentration range from 0 to 300 mg L^−1^. After shacking, the solid adsorbent and the liquid phase were separated by centrifugation.

All experiments were performed in replicate, and blank tests were realized. In addition, NdIII) quantification was performed by inductively coupled plasma optical emission spectrometry (ICP–OES) [36] (Perkin-Elmer, Waltham, MA, USA). The results were expressed in removal percentage and adsorption capacity (see Appendix A).

### 2.3. Kinetics, Equilibrium, and Thermodynamic Evaluation

Kinetics, equilibrium, and thermodynamic studies evaluated the Nd(III) biosorption on LEB-18 and LEB-52 *Spirulina platensis* algae. Further details are presented in the Appendix A [19,37,38].

## 3. Results and Discussion

### 3.1. SEM Images and Textural Properties

The images LEB-18 and LEB-52 *Spirulina platensis* algae are depicted in Figure 1. The difference between the strains of *Spirulina platensis* algae is distinct in this figure.

The strain LEB-18 of *S. platensis* is formed of small particles presenting more cavities between the aggregate of particles. On the other hand, the LEB-52 of *S. platensis* is denser, presenting fewer cavities between the particles. The LEB-18 strain presents more channels for solvent passage that will allow faster kinetics of uptake of Nd(III) species. It is important to highlight that SEM images after the adsorption are not important for being taken because this analytical technique has the ability to register images at µm scale, and the uptake of Nd^3+^ occurs at Å scale. Therefore, the papers that report the SEM images after the adsorption show the effect of the friction of the solvent on the adsorbent after the adsorption and not the uptaken specie retained in the adsorbent [39,40,41].

The surface area and total pore volume of LEB-18 *S. platensis* and LEB-52 *S. platensis* biomasses are presented in Table 1.

The obtained surface area and total pore volume of biomasses are compatible with previous data reported in the literature [27,28,29,30]. Both *Spirulina platensis* present low surface area and total pore volume, indicating that the main mechanism of Nd^3+^ uptake should not be pore filling.

### 3.2. FTIR

The FTIR of LEB-18 *S. platensis* (Figure 2a) and LEB-52 *S. Platensis* (Figure 2b) is presented in Figure 2.

The FTIR bands of these two strains are practically the same. The band at 3280 cm^−1^ (LEB-18) and 3278 cm^−1^ (LEB-52) are assigned to the stretching of O-H groups [42,43]. The bands at 2959 and 2920 cm^−1^ (LEB18) or 2921 cm^−1^ (LEB-52) are assigned to asymmetric C-H stretching. In addition, the band at 2851 cm^−1^ is attributed to symmetric C-H stretching [42,43]. The band at 1634 cm^−1^ could be assigned to C=O present in lipids of *S. platensis* [42,43]. The strong band at 1541 cm^−1^ is assigned to the N-H bending of amide groups present in alga [42,43]. The shoulder at 1449 (LEB-18) and 1452 cm^−1^ (LEB-52) and the band at 1400 cm^−1^ could be assigned to sulfates present in algae [42,43]. The bands at 1313 (LEB-18) and 1315 cm^−1^ (LEB-52) are assigned to the C-N of amines or amides [42,43]. The band at 1047 cm^−1^ (LEB-18) and 1039 cm^−1^ (LEB-52) could be assigned to the C-O stretching of carbohydrates or lipids as well as to P-O bonds of phosphates groups of algae [42,43]. The bands at 867 (LEB-18) and 872 cm^−1^ (LEB-52) are assigned to NH_2_ bending [42,43].

It is noteworthy to report that FTIR analytical technique has not enough sensitivity to detect an adsorbate uptaken by an adsorbent, although this is a common presentation of the FTIR data in the literature [44,45,46,47,48,49,50]. Furthermore, the usual resolution of FTIR equipment is 4 cm^−1^; therefore, any band shift < 12 cm^−1^ could not be assigned to any bond between the adsorbate and the adsorbent. Although it is a common practice in different papers dealing with adsorption, performing FTIR analysis after adsorption is not recommended [51]. Another important point is the decrease in FTIR band intensities discussed in most papers dealing with the adsorption of adsorbate in an adsorbent using FTIR spectra after the adsorption [51]. Usually, the authors use KBr pellets. Each pellet has a different optical path, even using the same experimental conditions (mass of KBr, pressure of the pastillator). Therefore, any comment made on band intensity has no physical meaning [51]; however, most authors neglect these remarkable observations [51].

### 3.3. Elemental Composition

The C, N, H, O, P, and S percentages of both biomasses are presented in Table 2.

Of course, both biomasses are composed of C, N, H, O, P, and S. The main difference is that the LEB-18 strain contains more P and S elements. Otherwise, the LEB-52 contains a higher amount of N. These differences could be attributed to the cultivation forms and strains. The biomasses’ elements N, P, and S are arranged as amines and amides, phosphates, and sulfates. These groups, in turn, affect the surface charge of the biosorbents due to protonation or deprotonation, leading to different biosorption behaviors.

### 3.4. pH Effect on the Nd Biosorption and pH_pzc_

Figure 3 shows the pH effect on the Nd biosorption by LEB-18 and LEB-52 strains. This pH range was selected since a pH higher than 7.0 Nd starts to precipitate.

Figure 3 demonstrates two facts. The first is that for both strains, the biosorption increased with the pH. For LEB-52, the increase was from 5% to 77.5%, and for LEB-18, the removal percentage attained 98.5%. This trend is related to the competition of Nd^3+^ positively charged with the H^+^ ions in the solution. There is lower competition as the pH increase, leading to higher values of Nd^3+^ removal. The second fact is that regardless of the pH, LEB-18 presented better results for Nd^3+^ removal than the LEB-52 strain. This trend could be attributed to the differences between the surface groups of the biomasses. LEB-52 contains more N (Table 1), arranged in NH_2,_ and easily converted to NH_3_^+^, repealing the Nd^3+^-positive ions. On the other hand, LEB-18 contains more P and S (Table 1), which forms PO_4_^3−^ and SO_4_^2−^ attracting the Nd^3+^ or complexing it [27,28,30].

Also, the pHpzc of LEB-18 and LEB-52 confirms these results (see Figure 4). The pHpzc of LEB-18 (5.62) and LEB-52 (5.81) agree with the pH studies. At pH 6.0, both biosorbents present a negatively charged surface with a high tendency to attract cations such as Nd^3+^.

### 3.5. Kinetics of Nd(II) Biosorption

The kinetics of biosorption of Nd(III) onto LEB-18 *S. platensis* and LEB-52 *S. platensis* was carried out utilizing the pseudo-first-order (PFO), pseudo-second-order (PSO), and Avrami-fractional-order kinetic models. The data are displayed in Figure 5.

The values of the kinetic parameters are depicted in Table 3. The statistical evaluation of the kinetic models was based on R^2^_adj_, SD, and BIC values. The best kinetic fitted model would present R^2^_adj_ closer to 1, with lower values of SD and BIC. However, the difference between the BIC values of the two models has values to be considered [19,37]. When ΔBIC < 2, there is no statistical difference between the two models; 2 < ΔBIC < 6, there is a tendency for the model with lower BIC values to be the most suitable model; 6 < ΔBIC < 10, the model with lower BIC value, has a strong possibility of being the best-fitted model, ΔBIC > 10 is certainly the best-fitted model.

For LEB-18 *S. platensis* biosorbent, the ΔBIC values between Avrami and PFO and Avrami and PSO were 7.71 and 10.9, respectively, for 20.0 mg L^−1^ of Nd(III) initial concentration and the ΔBIC values between Avrami and PFO and Avrami and PSO were 9.14 and 36.2, respectively for 50.0 mg L^−1^ of Nd(III). These results show a strong possibility that Avrami kinetic model can suitably explain the adsorption kinetic of Nd(III) onto LEB-18 *S. platensis* biosorbent.

For LEB-52 *S. platensis* biosorbent, the ΔBIC was: 20.0 mg L^−1^ for PFO-Avrami 14.6 and PSO-Avrami 38.2, and for 50.0 mg L^−1^, the ΔBIC was PFO-Avrami 2.57 and Avrami-PSO 35.9. Although the Avrami was the best kinetic model for the initial concentration of 20.0 mg L^−1^, the PFO presented a slighter advantage for 50.0 mg L^−1^. The Avrami-fractional order was a suitable model to describe the kinetics of Nd(III) adsorption onto the LEB-18 strain; for the LEB-52 strain using 50.0 mg L^−1^, the PFO kinetic model has a small advantage over Avrami.

The Avrami-fractional kinetic model was largely utilized to describe the kinetic of different adsorbates uptaken for different adsorbents. This model is associated with the changes in the order of the kinetics (n_AV_) during the contact of the adsorbate with the adsorbent [19].

The t_1/2_ and t_0.95_ are defined as the time to attain 50% and 95% of the saturation of the adsorbent [19,52]. Considering that Avrami fractional order was the best-fitted model, it could be established that the values of these time parameters were more confident than the other models. Therefore, it could be considered that 80 min would be a time necessary to attain the equilibrium of Nd(III) using both adsorbents, considering the maximum t_0.95_ values of 65.36 (LEB-18) and 69.80 min (LEB-52). Usually, the t_eq_ > t_0.95_ guarantees that the time of contact between the adsorbent and adsorbate is enough to attain equilibrium.

### 3.6. Equilibrium and Thermodynamics and Mechanism

Nd(III) biosorption equilibrium experiments on LEB-18 *Spirulina platensis* and LEB-52 *Spirulina platensis* algae were carried out from 25 to 55 °C (Table 4). The equilibrium isotherm models used were Langmuir, Freundlich, and Liu. These isotherm models were evaluated using R^2^, R^2^_adj_, SD, and BIC values (Table 4). For the set of four temperatures and two biosorbents, the Liu model was the best isotherm model for describing the uptake of Nd(III) onto both biosorbents. Except for LEB-18 at 298K, all values of ΔBIC between Langmuir and Liu and Freundlich and Liu were > 10, indicating that surely the Liu isotherm model was the best isotherm for describing the adsorption of Nd(III) onto LEB-18 *Spirulina platensis* and LEB-52 *Spirulina platensis* algae [19,38,52]. For the LEB-18 strain at 298 K, the ΔBIC was 5.74, indicating that the Liu isotherm model might be the best choice of the isotherm model [19,38,52]. The R^2^_adj_ and SD values of the Liu isotherm model were also closer to 1 and minimum values, respectively, confirming that the Liu isotherm model was the best choice.

The maximum biosorption capacities were 72.5 mg g^−1^ for LEB-18 and 48.2 mg g^−1^ for LEB-52 at a temperature of 298 K. These values are competitive with other materials used to uptake Nd from aqueous solutions. For example, Javadian et al. [53] compared around 20 adsorbents used to uptake Nd from aqueous matrices. They found capacity values from 27.1 to 126.5 mg g^−1^. Moreover, the biosorbents were tested in real samples of phosphogypsum leachate [5]. This leachate is an H_2_SO_4_ solution containing 183 mg L^−1^ of Ce, 95.7 mg L^−1^ of Nd, 83 mg L^−1^ of La, 12.7 mg L^−1^ of Sm, and other rare earth elements at concentrations lower than 10 mg L^−1^. The biosorbents were efficient even under acid conditions removing more than 80% f Nd of the leachate.

Figure 5 shows at left the isotherms of adsorption of Nd(III) onto LEB-18 strain (Figure 5a) and LEB-52 strain (Figure 5b) at 25 °C. On the right, Figure 5c (LEB-18) and Figure 5d (LEB-52) show the nonlinear Van’t Hoff curves for the determination of the thermodynamic parameters of adsorption of Nd(III) onto both biosorbents [19,38,52].

The values of ΔH° and ΔS° of adsorption are depicted in Table 5 and Figure 6c,d. The $K_{e}^{0}$ was obtained as previously recommended [19,38,52] based on the best-fitted isotherm model (Liu), see Appendix A. According to the thermodynamic data, the adsorption process was favorable (ΔG° < 0 for all the studied temperatures) and exothermic (ΔH° < 0). The changes in the entropy were positive, indicating that Nd(III) adsorbed in the biosorbent surface is at a more organized state compared with it dissolved in the bulk of the solution.

Based on the results of the characterization of LEB-18 and LEB-52 of *Spirulina platensis* biosorbents and the kinetics, equilibrium, and thermodynamic data is possible to propose a mechanism of adsorption. As both biosorbents present low surface area and total pore volume, the pore-filling mechanism is ruled out. Furthermore, the pH_pzc_ of the biosorbents and the initial pH studies suggest that at pH 6.0 (see Figure 3 and Figure 4), the phosphate and sulfate groups of both biosorbents (see Table 2) are unprotonated, being available for interacting with Nd^3+^ species by electrostatic attraction (whose value of ΔH° adsorption is compatible with this physical interaction, see Figure 6 and Table 5). The higher sorption capacity of the LEB-18 strain over LEB-52 is based on the amount of PO_4_^3−^ and SO_4_^2−^ moieties present in the LEB-18 strain (see Table 4). Based on these explanations, a schematic diagram of the adsorption mechanism is shown in Figure 7.

## 4. Conclusions

In this study, two strains of *Spirulina platensis* (LEB-18 and LEB-52) were used as biosorbents for neodymium in an aqueous solution. The biosorbents were characterized and further applied in biosorption experiments. The biosorption was favored at pH 6, and the LEB-18 strain presented a better potential to uptake the Nd. The better potential of the LEB-18 strain could be related to the higher P and S content in this composition. P and S are arranged as phosphates and sulfates that could uptake Nd.

Concerning the kinetics, the Avrami model best represented the experimental data of Nd adsorption for the two biomasses based on R^2^_adj_, SD, and BIC values. The values of t_0.95_, which indicate the time to attain 95% of the saturation of the biosorbent, showed that the kinetics of LEB-18 needs a shorter time to come to equilibrium than that of LEB-52, indicating a higher affinity for the LEB-18/Nd system. This fact is supported by the theoretical biosorption capacities found in the Avrami model for both Nd concentrations used, being higher for LEB-18 biomass than LEB-52 biomass. This trend also corroborates the results found in biomass characterization, especially concerning scanning electron microscopy, where LEB18 particles have more cavities than LEB52 particles.

The Liu model for both biosorbents better described the biosorption isotherm curves. The higher theoretical maximum capacity values reported by Liu’s model were found for LEB- 18 biomass (72.45 mg g^−1^ at 298 K), compared to 42.24 mg g^−1^ found for LEB-52 biomass. The biosorption was exothermic for the two spirulina biomasses. In addition, it is important to highlight that temperatures higher than 40 °C can already initiate the degradation of the *Spirulina* functional groups. In addition to the results found in elemental analysis, the greater efficiency of LEB-18 concerning LEB-52 may be due to a greater presence of negative phosphate and sulfate groups, which can interact with positive neodymium, facilitating electrostatic attraction, and consequently increasing the biosorption capacity of the material.

## Figures and Tables

**Figure 1 polymers-14-04585-f001:**
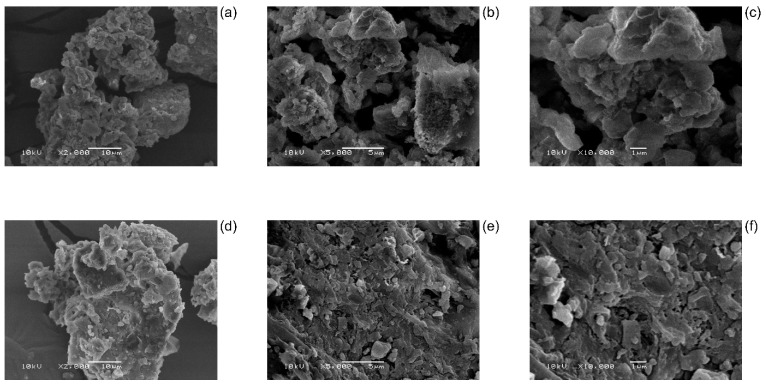
SEM images of LEB-18 (**a**–**c**) and LEB-52 (**d**–**f**) *S. platensis*. Augmentation factor 2000× (**a**,**d**), 5000× (**b**,**e**), and 10,000× (**c**,**f**).

**Figure 2 polymers-14-04585-f002:**
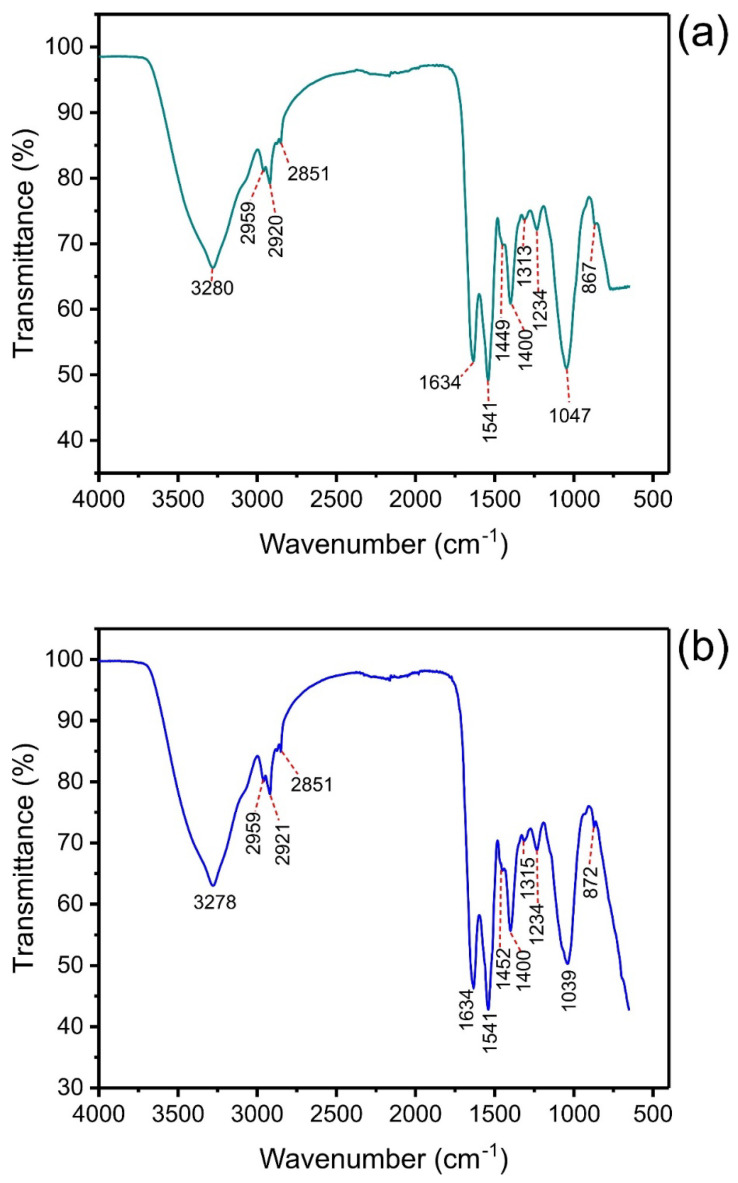
FTIR of (**a**) LEB-18 *S. platensis*; (**b**) LEB-52 *S. platensis*.

**Figure 3 polymers-14-04585-f003:**
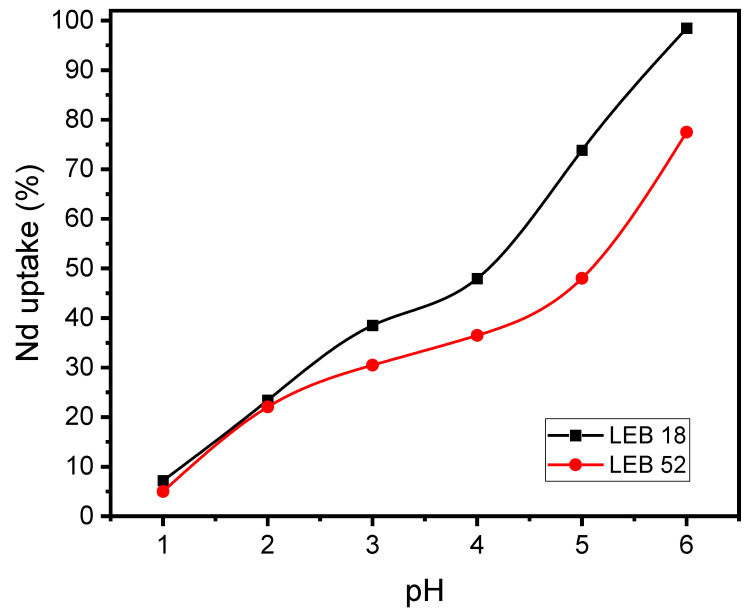
pH effect on the Nd biosorption by LEB-18 and LEB-52 *S. platensis*. Initial concentration 20.0 mg L^−1^, temperature 25 °C, adsorbent dosage 2.0 g L^−1^.

**Figure 4 polymers-14-04585-f004:**
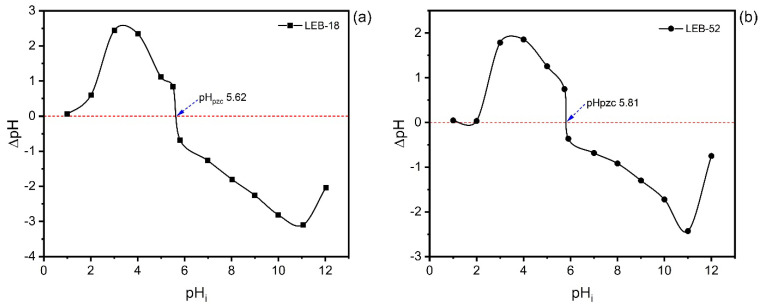
pHpzc of LEB-18 (**a**) and LEB-52 (**b**) biosorbents.

**Figure 5 polymers-14-04585-f005:**
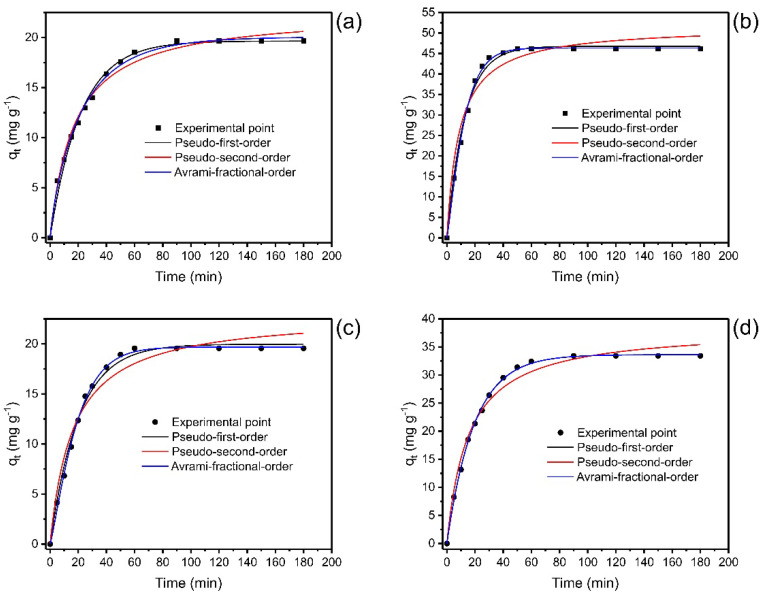
Kinetics curves for the biosorption of Nd(III) onto LEB-18 *S. platensis* (**a**,**b**) and LEB-52 *S. platensis* (**c**,**d**). Initial concentration 20.0 mg L^−1^ (**a**,**c**) and 50.0 mg L^−1^ (**b**,**d**). Conditions: initial pH 6.0; temperature 25 °C; adsorbent dosage 2 g L^−1^.

**Figure 6 polymers-14-04585-f006:**
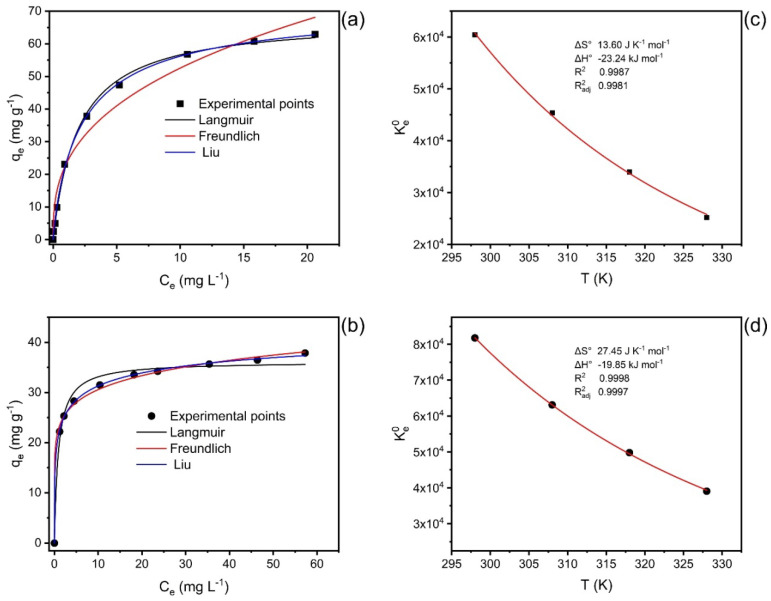
Isotherms at 298 K (**a**,**b**) and nonlinear plot of Van’t Hoff (**c**,**d**) for the uptake of Nd(III) onto LEB-18 (**a**,**c**) and LEB-52 (**b**,**d**) *S. platensis* biosorbent.

**Figure 7 polymers-14-04585-f007:**
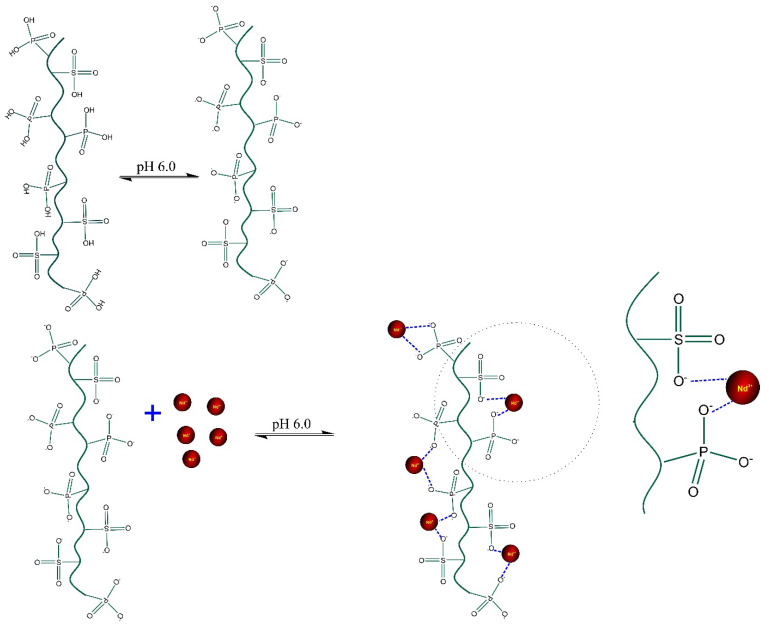
Proposed mechanism of adsorption of Nd^3+^ by both LEB-18 and LEB-52 *Spirulina platensis* biosorbents.

**Table 1 polymers-14-04585-t001:** Textural properties of LEB-18 *S. platensis* and LEB-52 *S. platensis* biomasses.

	LEB-18 *S. platensis*	LEB-52 *S. platensis*
Surface area (m^2^ g^−1^)	4.65	3.63
Total pore volume (cm^3^ g^−1^)	0.0060	0.0040

**Table 2 polymers-14-04585-t002:** Elemental composition of LEB-18 *S. platensis* and LEB-52 *S. platensis* biomasses.

Element	LEB-18 * (%)	LEB-52 * (%)
C	44.2 ± 2.1	45.7 ± 1.5
H	11.2 ± 0.3	10.7 ± 0.2
N	24.3 ± 1.0	31.3 ± 1.5
O	9.1 ± 0.5	9.3 ± 0.2
P	6.7 ± 0.1	1.7 ± 0.1
S	4.5 ± 0.1	1.3 ± 0.1

* Mean ± standard error for n = 3.

**Table 3 polymers-14-04585-t003:** Kinetic parameters for Nd(III) adsorption onto LEB-18 *S. platensis* and LEB-52 *S. platensis*. Conditions: temperature 25 °C, adsorbent dosage 2.0 g L^−1^, pH 6, initial adsorbate concentration 20.0, and 50.0 mg L^−1^.

	C_o_ 20.0 mg L^−1^		C_o_ 50.0 mg L^−1^	
Avrami	LEB-18	LEB-52	LEB-18	LEB-52
q_e_ (mg g^−1^)	20.06	19.68	46.33	33.63
k_AV_ (min^−1^)	0.04429	0.05049	0.07803	0.05144
n_AV_	0.8500	1.1899	1.174	0.9944
t_1/2_ (min)	14.60	14.56	9.377	13.45
t_0.95_ (min)	65.36	69.80	32.64	58.56
R^2^ adjusted	0.9953	0.9978	0.9971	0.9988
SD (mg g^−1^)	0.4233	0.3067	0.7847	0.3723
BIC	−16.89	−25.91	0.3930	−20.48
PFO				
q_e_ (mg g^−1^)	19.65	19.97	46.76	33.61
k_1_ (min^−1^)	0.04601	0.04946	0.07841	0.05148
t_1/2_ (min)	15.06	14.01	8.840	13.46
t_0.95_ (min)	65.00	60.51	38.20	58.16
R^2^_adj_	0.9909	0.9932	0.9938	0.9989
SD (mg g^−1^)	0.5864	0.5428	1.144	0.3573
BIC	−9.187	−11.35	9.534	−23.06
PSO				
q_e_ (mg g^−1^)	22.59	22.99	51.69	38.41
k_2_ (g mg^−1^ min^−1^)	0.002557	0.002637	0.002188	0.001700
t_1/2_ (min)	14.52	13.94	8.052	13.09
t_0.95_ (min)	112.5	110.6	84.75	107.7
R^2^_adj_	0.9886	0.9630	0.9570	0.9826
SD (mg g^−1^)	0.6565	1.263	3.008	1.413
BIC	−6.024	12.30	36.60	15.44

**Table 4 polymers-14-04585-t004:** Biosorption parameters isotherm models for removing Nd(III) onto LEB-18 and LEB-52 *Spirulina platensis* biosorbents.

LEB-18	298 K	308 K	318 K	328 K
Langmuir				
Q_max_ (mg g^−1^)	67.81	43.78	34.74	21.52
K_L_ (L mg^−1^)	0.5055	3.665	0.2681	0.2749
R^2^_adj_	0.9975	0.9202	0.9980	0.9894
SD (mg g^−1^)	1.270	4.294	0.5160	0.7194
BIC	9.455	33.82	−8.588	−1.909
Freundlich				
K_F_ (mg g^−1^ (mg L^−1^)^−1/nF^)	22.99	28.77	12.72	8.855
n_F_	2.784	7.471	4.174	5.025
R^2^_adj_	0.9649	0.9950	0.9641	0.9883
SD (mg g^−1^)	4.777	1.079	2.203	0.7551
BIC	35.95	6.188	20.47	−0.9416
Liu				
Q_max_ (mg g^−1^)	72.45	69.64	36.59	25.38
K_g_ (L mg^−1^)	0.4189	0.3147	0.2353	0.1744
n_L_	0.8667	0.2768	0.8385	0.5879
R^2^_adj_	0.9987	0.9996	0.9999	0.9999
SD (mg g^−1^)	0.9081	0.2870	0.03009	0.01837
BIC	3.716	−19.32	−64.42	−74.30
LEB-52	298 K	308 K	318 K	328 K
Langmuir				
Q_max_ (mg g^−1^)	36.16	29.01	25.99	12.72
K_L_ (L mg^−1^)	1.092	0.6881	0.5099	0.3690
R^2^_adj_	0.9836	0.9848	0.9846	0.9862
SD (mg g^−1^)	1.437	1.141	1.014	0.4679
BIC	11.92	7.310	4.948	−10.51
Freundlich				
K_F_ (mg g^−1^ (mg L^−1^)^−1/nF^)	22.87	15.99	13.47	6.416
n_F_	7.931	6.754	6.155	6.469
R^2^_adj_	0.9967	0.9929	0.9882	0.9894
SD (mg g^−1^)	0.6407	0.7790	0.8877	0.4105
BIC	−4.228	−0.3179	2.295	−13.13
Liu				
Q_max_ (mg g^−1^)	48.24	35.39	31.08	15.03
K_g_ (L mg^−1^)	0.5669	0.4375	0.3451	0.2706
n_L_	0.3545	0.4642	0.5289	0.5285
R^2^_adj_	0.9993	0.9999	0.9999	0.9999
SD (mg g^−1^)	0.2897	0.003047	0.02917	0.02459
BIC	−19.14	−110.2	−65.05	−68.46

**Table 5 polymers-14-04585-t005:** Nd(III) adsorption thermodynamic parameters onto LEB-18 and LEB-52 biosorbents.

	Temperature (K)			
LEB-18	298	308	318	328
K_g_ (L mol^−1^)	6.042 × 0^4^	4.539 × 10^4^	3.394 × 10^4^	2.516 × 10^4^
$K_{e}^{0}$	6.042 × 10^4^	4.539 × 10^4^	3.394 × 10^4^	2.516 × 10^4^
∆G° (kJ mol^−1^)	−27.28	−27.46	−27.58	−27.63
∆H° (kJ mol^−1^)	−23.24	-	-	-
∆S° (J K^−1^ mol^−1^)	13.60	-	-	-
R^2^_adj_	0.9981			
LEB-52	298	308	318	328
K_g_ (L mol^−1^)	8.177 × 10^4^	6.311 × 10^4^	4.978 × 10^4^	3.903 × 10^4^
$K_{e}^{0}$	8.177 × 10^4^	6.311 × 10^4^	4.978 × 10^4^	3.903 × 10^4^
∆G° (kJ mol^−1^)	8.177 × 10^4^	6.311 × 10^4^	4.978 × 10^4^	3.903 × 10^4^
∆H° (kJ mol^−1^)	−19.85	-	-	-
∆S° (J K^−1^ mol^−1^)	27.45	-	-	-
R^2^_adj_	0.9997			

## Data Availability

The data are available on request.

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
