# Peer review of "Biosorption of Neodymium (Nd) from Aqueous Solutions Using Spirulina platensis sp. Strains"

_polymers, 2022, doi:10.3390/polym14214585_

Round 1

Reviewer 1 Report

Dear Mr. Lionel Lin

MDPI Polymers Editorial Office

Thank you for choosing me to review the manuscript entitled: “Biosorption of Neodymium (Nd) from aqueous effluents using Spirulina platensis sp. strains” by Lima et al.

- This article is primarily valuable for considering heavy metal pollution (Neodymium) so in my opinion the authors have written a useful and resourceful study for pollution of environmental. However, I have some comments for improvement the current manuscript before resubmission the manuscript to Polymers.

General comments

- In introduction section: Although REEs have been uptaken …………………….. to report the Nd uptake using Spirulina platensis (green micro-algae).

Spirulina belongs to blue-green algae not green.

- In the materials and methods section, it is necessary to mention the percentage of purity for Neodymium nitrate hexahydrate.

- Authors have choose only one factor including pH, Why did use this factor? They could choose some significant factors such as temperature, sample volume, adsorbent dosage, initial adsorbate concentration, shaker ratio or contact time as independent factors for studying their effect on the removal percentage of Nd. Why was the initial concentration of 50 mg L-1 chosen in the pH study? What can they say about it?

- What do you mean by "each chitosan-based material was inserted in Nd(III) solutions" in the materials and methods section?

- The biosorbent was characterized by SEM, EDX and FTIR techniques but they have not use Brauner Emmet- Teller (BET) analysis because the surface area of biosorbent is important in biosorption studies. In addition, there are no data regarding these techniques after biosorption process. It is necessary to add these data and images of the strains after biosorption.    

- The authors did not explain pHpzc clearly in the manuscript. Although the method is mentioned in Materials and methods, it is recommended to supplement the manuscript with data and a graph of pH change.

- At part pH, it is necessary to add information about the importance and necessity of the pH parameter. Also, the pH parameter should be interpreted according to pHzpc. 

It is necessary to refer to the following article:

https://doi.org/10.1038/s41598-022-07288-z

- The results must be compared with other studies for deeper digging.

- The explanation of the possible mechanism of Nd(III) biosorption should be added in the manuscript.

- The obvious problem with this manuscript is the lack of sufficient experimentation to demonstrate the validity and applicability of the proposed method. Too few experiments make the conclusion of this paper lack persuasive. 

- Some related research can be cited for deeply discussed, which would be beneficial for paper quality.

https://doi.org/10.1016/j.jenvman.2020.110380

https://doi.org/10.1016/j.eti.2020.100953

- According to these results, has it been tested in real sewage? If you want to work on a real scale, what is the effect of the presence of other interfering factors on the efficiency of the process? How about economic efficiency?

- I recommended that this paper be accepted for publication in this journal after major revision.

Author Response

General comments

1) - In the introduction section: Although REEs have been uptaken …………………….. to report the Nd uptake using Spirulina platensis (green micro-algae). Spirulina belongs to blue-green algae, not green.

In the revised manuscript: "Although REEs have been uptaken utilizing biosorbents, up to the best of our knowledge, this paper is the first to report the Nd uptake using Spirulina platensis (blue-green micro-algae)."

2) In the materials and methods section, it is necessary to mention the percentage of purity for Neodymium nitrate hexahydrate.

In the revised manuscript: "Neodymium nitrate hexahydrate (Nd(NO₃)₃.6Hâ‚‚O; 99.9% purity)…"

3) Authors have chosen only one factor, including pH; why did they use this factor? They could choose some significant factors such as temperature, sample volume, adsorbent dosage, initial adsorbate concentration, shaker ratio, or contact time as independent factors for studying their effect on the removal percentage of Nd. Why was the initial concentration of 50 mg L-1 chosen in the pH study? What can they say about it?

Dear author, the pH of the initial adsorbate concentration is paramount because precipitation and formation of complexes with hydroxyl could occur. Besides that, the other parameters, such as time of contact, are explored in the kinetic experiments; the effect of initial concentration is studied in the isotherms of adsorption; the effect of temperature is explored in the thermodynamic studies. The effect of shaker speed depends on each equipment model, which is usually optimized when the equipment is purchased. Therefore it is not relevant to define its values because the best condition of one piece of equipment will be different in another shaker type.

We tested 50 mg L-1 of Nd at different pH values to start the adsorption experiments, but this was an experimental try, and it would be possible to have used a lower initial concentration of the adsorbate to attain these studies.

4) What do you mean by "each chitosan-based material was inserted in Nd(III) solutions" in the materials and methods section?

The revised manuscript: "In these experiments, each microalgae material was inserted in Nd(III) solutions…."

5) The biosorbent was characterized by SEM, EDX, and FTIR techniques, but they have not used Brauner Emmet-Teller (BET) analysis because the surface area of the biosorbent is important in biosorption studies. In addition, there are no data regarding these techniques after the biosorption process. It is necessary to add these data and images of the strains after biosorption.    

The revised manuscript: "The surface area and total pore volume of LEB-18 S. platensis and LEB-52 S. platensis biomasses are presented in Table 1.

Table 1. Textural properties of LEB-18 S. platensis and LEB-52 S. platensis biomasses

LEB-18 S. platensis

LEB-52 S. platensis

Surface area (m2 g-1)

4.65

3.63

Total pore volume (cm3 g-1)

0.0060

0.0040

            The obtained surface area and total pore volume of biomasses are compatible with previous data reported in the literature [27-30]. Both Spirulina platensis present low surface area and total pore volume, indicating that the main mechanism of Nd3+ uptake should not be pore filling."

6)- The authors did not explain pHpzc clearly in the manuscript. Although the method is mentioned in Materials and methods, it is recommended to supplement the manuscript with data and a graph of pH change.

In the revised manuscript:

"LEB-18 presented better results for Nd3+ removal than LEB-52 strain. This trend could be attributed to the differences between the surface groups of the biomasses. LEB-52 contains more N (Table 1), arranged in NH2 and easily converted to NH3+, repealing the Nd3+ positive ions. On the other hand, LEB-18 contains more P and S (Table 1), which forms PO43- and SO42- attracting the Nd3+ or complexing it.

Also, the pHpzc of LEB-18 and LEB-52 confirms these results (see Fig 4). The pHpzc of LEB-18 (5.62) and LEB-52 (5.81) agree with the pH studies. At pH 6.0, both biosorbents present a negatively-charged surface with a high tendency to attract cations such as Nd3+.

Fig 4. pHpzc of LEB-18 (a) and LEB-52 (b) biosorbents."

7)- At part pH, it is necessary to add information about the importance and necessity of the pH parameter. Also, the pH parameter should be interpreted according to pHzpc. 

It is necessary to refer to the following article: https://doi.org/10.1038/s41598-022-07288-z

REF 42 was inserted in the revised manuscript

[42] M.A. Fawzy, H. Darwish, S. Alharthi, M.I. Al‑Zaban, A. Noureldeen, S.H.A. Hassan. Process optimization and modeling of Cd2+ biosorption onto the free and immobilized Turbinaria ornata using Box–Behnken experimental design. Scientific Reports 12,(2022) 3256. Doi:10.1038/s41598-022-07288-z.

The importance of pH and pHpzc were presented in the previous query.

8) The results must be compared with other studies for deeper digging.

See in the revised manuscript: "The maximum biosorption capacities were 72.5 mg g-1 for LEB 18, and 48.2 mg g-1 for LEB 52 at a temperature of 298 K. These values are competitive with other materials used to uptake Nd from aqueous solutions. For example, Javadian et al. [53] compared around 20 adsorbents used to uptake Nd from aqueous matrices. They found capacity values from 27.1 to 126.5 mg g-1."

REF [53] was added:

[53] H Javadian, M. Ruiz, M. Taghvai, A.M. Saestre. Novel magnetic nanocomposite of calcium alginate carrying poly(pyrimidine-thiophene-amide) as a novel green synthesized polyamide for adsorption study of neodymium, terbium, and dysprosium rare-earth ions. Colloids and Surfaces A: Physicochemical and Engineering Aspects 603 (2020) 125252. DOI: 10.1016/j.colsurfa.2020.125252.

9) The explanation of the possible mechanism of Nd(III) biosorption should be added to the manuscript.

In the revised manuscript:

"Based on the results of the characterization of LEB-18 and LEB-52 of Spirulina platensis biosorbents and the kinetics, equilibrium, and thermodynamic data is possible to propose a mechanism of adsorption. As both biosorbents present low surface area and total pore volume, the pore filling mechanism is ruled out. Furthermore, the pHpzc of the biosorbents and the initial pH studies suggest that at pH 6.0 (see Fig 3 and Fig 4), the phosphate and sulfate groups of both biosorbents (see Table 2) are unprotonated, being available for interacting with Nd3+ species by electrostatic attraction (whose value of DH° adsorption is compatible with this physical interaction, see Fig 6 and Table 5). The higher sorption capacity of the LEB-18 strain over LEB-52 is based on the amount of PO43- and SO42- moieties present in the LEB-18 strain (see Table 4). Based on these explanations, a schematic diagram of the adsorption mechanism is shown in Fig 7."

Fig 7. Proposed mechanism of adsorption of Nd3+ by both LEB-18 and LEB-52 Spirulina platensis biosorbents

10) The obvious problem with this manuscript is the lack of sufficient experimentation to demonstrate the validity and applicability of the proposed method. Too few experiments make the conclusion of this paper lack persuasive. 

- Some related research can be cited for deeply discussed, which would be beneficial for paper quality.

https://doi.org/10.1016/j.jenvman.2020.110380

https://doi.org/10.1016/j.eti.2020.100953

The following reference was added in the revised manuscript:

[41] M.A. Fawzy, H. Darwish, S. Alharthi, M.I. Al‑Zaban, A. Noureldeen, S.H.A. Hassan. Process optimization and modeling of Cd2+ biosorption onto the free and immobilized Turbinaria ornata using Box–Behnken experimental design. Scientific Reports 12,(2022) 3256. Doi:10.1038/s41598-022-07288-z.

[43] M.A.Fawzy, M. Gomaa. Use of algal biorefinery waste and waste office paper in the development of xerogels: A low-cost and eco-friendly biosorbent for the effective removal of congo red and Fe (II) from aqueous solutions. Journal of Environmental Management, 262 (2020) 110380. Doi:10.1016/j.jenvman.2020.110380.

[46] M.A. Fawzy, A.F.Hifney, M.S. Adam, A.A.Al-Badaani. Biosorption of cobalt and its effect on growth and metabolites of Synechocystis pevalekii and Scenedesmus bernardii: Isothermal analysis. Environmental Technology & Innovation 19 (2020) 100953. Doi: 10.1016/j.eti.2020.100953.

         The experimental results included the pHpzc, the textural data of the surface area, and the total pore volume. In addition, the mechanism of adsorption was also proposed. All the results of equilibrium and kinetic results were performed using nonlinear fitting that is more coherent than the linear approach usually employed in the majority of papers reported in the literature. The errors of using linearization have been extensively discussed in references that we have cited in this current paper:

[19] E.C. Lima, M.H. Dehghani, A. Guleria, F. Sher, R.R. Karri, G.L. Dotto, H.N. Tran, CHAPTER 3 - Adsorption: Fundamental aspects and applications of adsorption for effluent treatment, in: Hadi Dehghani, M., Karri, R., Lima, E. (Eds.), Green Technologies for the Defluoridation of Water. Elsevier, 2021, 41–88. Doi:10.1016/B978-0-323-85768-0.00004-X.

[37] E.C. Lima, F. Sher, A. Guleria, M.R. Saeb, I. Anastopoulos, H.N. Tran, A. Hosseini-Bandegharaei, Is one performing the treatment data of adsorption kinetics correctly? J Environ Chem Eng 9 (2021) 104813. doi:10.1016/j.jece.2020.104813.

In addition, the thermodynamic results were correctly obtained using the best-fitted isotherm data at four different temperatures, which is quite different from the common papers dealing with thermodynamics that use Kc or Kd instead of the thermodynamic equilibrium constant. Please see the references below:

[19] E.C. Lima, M.H. Dehghani, A. Guleria, F. Sher, R.R. Karri, G.L. Dotto, H.N. Tran, CHAPTER 3 - Adsorption: Fundamental aspects and applications of adsorption for effluent treatment, in: Hadi Dehghani, M., Karri, R., Lima, E. (Eds.), Green Technologies for the Defluoridation of Water. Elsevier, 2021, 41–88. Doi:10.1016/B978-0-323-85768-0.00004-X.

[38] E.C. Lima, A. Hosseini-Bandegharaei, J.C. Moreno-Piraján, I. Anastopoulos. A critical review of the estimation of the thermodynamic parameters on adsorption equilibria. Wrong use of equilibrium constant in the Van't Hoof equation for calculation of thermodynamic parameters of adsorption. J Mol Liq 273 (2019) 425-434. Doi: 10.1016/j.molliq.2018.10.048.

11) According to these results, has it been tested in real sewage? If you want to work on a real scale, what is the effect of the presence of other interfering factors on the efficiency of the process? How about economic efficiency?

We agree. See in the revised manuscript:

"Besides, the biosorbents were tested in real samples of phosphogypsum leachate [54]. This leachate is an H2SO4 solution containing 183 mg L-1 of Ce, 95.7 mg L-1 of Nd, 83 mg L-1 of La, 12.7 mg L-1 of Sm, and other rare earth elements at concentrations lower than 10 mg L-1. The biosorbents were efficient even under acid conditions removing more than 80% f Nd of the leachate."

REF 54 was added:

[54] SF Lütke, MLS Oliveira, SR Waechter, LFO Silva, TRS Cadaval, FA Duarte, GL Dotto. Leaching of rare earth elements from phosphogypsum. Chemosphere 301 (2022) 134661. DOI: 10.1016/j.chemosphere.2022.134661.

Reviewer 2 Report

Comments regarding the manuscript entitled “Biosorption of Neodymium (Nd) from aqueous effluents using Spirulina platensis sp. strains”

Considering the interest that resides in the possible utilization of natural materials for the recovery of valuable elements from liquid solutions, new information on this topic is welcome. In this context, the paper's subject is interesting. However, some aspects require further attention before a conclusion can be made and the paper be accepted. My principal comments are as follows:

1)    Title: Biosorption experiments were carried out using neodymium nitrate hexahydrate solutions, not aqueous effluents as stated in the paper title. It would be better to say “aqueous solutions” to avoid misunderstandings.

2)    The authors stated that Spirulina platensis is a blue-green microalgae. However, Spirulina is a cyanobacteria. Although cyanobacteria have traditionally been classified as algae, referenced as cyanophytes or blue-green algae, today many treatises exclude them from algae.

3)    Authors did not maintain constant the pH of solution during Nd biosorption experiments. However, the change of the pH of solution during an adsorption experiment may lead to serious deviation in experimental results.

4)    SEM images of Nd-loaded S. platensis strains were not registered. Why? They could provide valuable information.

5)    Why were the FTIR spectra of the Nd-loaded Spirulina platensis strains not obtained? They could provide information on the functional groups responsible for biosorption and confirm or rule out the conclusions drawn by the authors.

6)    The authors stated that the C, N, H, O, P, and S percentages are presented in Table 1. However, the H percentage is missing in Table 1.

7)    The authors stated in section 2.2 that the fixed conditions of the biosorption experiments were: adsorbent dosage of 1 g/L, solution volume of 50 mL, and stirring rate of 200 rpm. However, in figures 3 and 4 it is mentioned that the biosorption experiments were carried out with an adsorbent concentration of 2 g/L.

8)    It is mentioned in section 2.2 that the assays of the effect of solution pH on Nd biosorption were carried out using an initial Nd concentration of 50 mg/L, but in the title of figure 3 it is mentioned that the initial concentration was 20 mg/L.

9)    In Fig. 4, the words “Pseudo-first order” are repeated.

10) Equilibrium time can be obtained by statistical analysis, why not estimate it this way? 

11) I suggest the comparison between other adsorbents used to remove Nd from aqueous solutions.

12) Please provide the activation energy for Nd biosorption.

13) Please check the Kg and Keº values for LEB-18 strain as well as the Kg, Keº and DGº values for LEB-52 strain. They are the same and this is wrong.

14) To recover Nd from Nd-loaded biosorbents, desorption studies are required. Please provide desorption data.

Author Response

12)    Title: Biosorption experiments were carried out using neodymium nitrate hexahydrate solutions, not aqueous effluents, as stated in the paper title. Therefore, it would be better to say "aqueous solutions" to avoid misunderstandings.

In the revised manuscript, the new title is:

Biosorption of Neodymium (Nd) from aqueous solutions using Spirulina platensis sp. strains

13)    The authors stated that Spirulina platensis is a blue-green microalga. However, Spirulina is a cyanobacteria. Although cyanobacteria have traditionally been classified as algae, referenced as cyanophytes or blue-green algae, today, many treatises exclude them from algae.

Dear reviewer, we used blue-green alga as recommended by the literature. We are not biologists to know if the definition is correct or not. This denomination will not change the results of this manuscript and will not be wrong because we have based on given denominations of the literature.

14)    Authors did not maintain the constant pH of the solution during Nd biosorption experiments. However, the change in the pH of the solution during an adsorption experiment may lead to serious deviation in experimental results.

The use of buffer solutions to keep the pH of the adsorbate constant in batch adsorption studies is not recommended because the ionic strength would affect the adsorption results. Besides that, buffering the pH of a real solution is hard to perform in a real application. We followed the traditional way that research on batch adsorption studies is carried out.

15)    SEM images of Nd-loaded S. platensis strains were not registered. Why? They could provide valuable information.

We explained in the revised manuscript:

"It is important to highlight that SEM images after the adsorption are not important for being taken because this analytical technique has the ability to register images at µm scale, and the uptake of Nd3+ occurs at Å scale. Therefore the papers that report the SEM images after the adsorption show the effect of the friction of the solvent on the adsorbent after the adsorption and not the uptaken specie retained in the adsorbent [39-41]."

16)    Why were the FTIR spectra of the Nd-loaded Spirulina platensis strains not obtained? They could provide information on the functional groups responsible for biosorption and confirm or rule out the conclusions drawn by the authors.

We explained in the revised manuscript:

"It is noteworthy to report that FTIR analytical technique has not enough sensitivity to detect an adsorbate uptaken by an adsorbent, although this is a common presentation of the FTIR data in the literature [44-50]. Furthermore, the usual resolution of FTIR equipment is 4 cm-1; therefore, any band shift < 12 cm-1 could not be assigned to any bond between the adsorbate and the adsorbent. Although it is a common practice in different papers dealing with adsorption, performing FTIR analysis after adsorption is not recommended [51]. Another important point is the decrease of FTIR band intensities discussed in most papers dealing with the adsorption of adsorbate in an adsorbent using FTIR spectra after the adsorption [51]. Usually, the authors use KBr pellets. Each pellet has a different optical path, even using the same experimental conditions (mass of KBr, pressure of the pastillator). Therefore, any comment made on band intensity has no physical meaning [51]; however, most authors neglect these remarkable observations [51]."

17)    The authors stated that the C, N, H, O, P, and S percentages are presented in Table 1. However, the H percentage is missing in Table 1.

In the revised manuscript:

Table 2. Elemental composition of LEB-18 S. platensis and LEB-52 S. platensis biomasses.

Element

LEB-18* (%)

LEB-52* (%)

C

44.2±2.1

45.7±1.5

H

11.2±0.3

10.7±0.2

N

24.3±1.0

31.3±1.5

O

9.1±0.5

9.3±0.2

P

6.7±0.1

1.7±0.1

S

4.5±0.1

1.3±0.1

*mean±standard error for n=3.

18)    The authors stated in section 2.2 that the fixed conditions of the biosorption experiments were: an adsorbent dosage of 1 g/L, solution volume of 50 mL, and stirring rate of 200 rpm. However, figures 3 and 4 mention that the biosorption experiments were carried out with an adsorbent concentration of 2 g/L.

This mistake was corrected in the revised manuscript: "The fixed conditions were: an adsorbent dosage of 2.00 g L-1, volume of the solution of 50 mL, and stirring rate of 200 rpm."

19)    It is mentioned in section 2.2 that the assays of the effect of solution pH on Nd biosorption were carried out using an initial Nd concentration of 50 mg/L, but in the title of figure 3, it is mentioned that the initial concentration was 20 mg/L.

In the revised manuscript:

"At first, the initial pH effect on Nd(III) adsorption was evaluated from 1.0 to 6.0. In these experiments, each microalgae material was inserted in Nd(III) solutions (20 mg L-1) and stirred for 2 h at 298 K."

20)    In Fig. 4, the words "Pseudo-first order" are repeated.

In the revised manuscript, this mistake was corrected.

21) Equilibrium time can be obtained by statistical analysis; why not estimate it this way? 

Based on values of t0.95 (time necessary to arrive at 95% saturation) it is possible to establish a minimum time to attain the equilibrium. In the revised manuscript:

"Considering that Avrami fractional order was the best-fitted model, it could be established that the values of these time parameters were more confident than the other models. Therefore, it could be considered that 80 min would be a time necessary to attain the equilibrium of Nd(III) using both adsorbents, considering the maximum t0.95 values of 65.36 (LEB-18) and 69.80 min (LEB-52). Usually, the teq > t0.95 guarantees that the time of contact between the adsorbent and adsorbate is enough for attaining equilibrium."

22) I suggest a comparison between other adsorbents used to remove Nd from aqueous solutions.

See in the revised manuscript: "The maximum biosorption capacities were 72.5 mg g-1 for LEB 18, and 48.2 mg g-1 for LEB 52 at a temperature of 298 K. These values are competitive with other materials used to uptake Nd from aqueous solutions. For example, Javadian et al. [53] compared around 20 adsorbents used to uptake Nd from aqueous matrices. They found capacity values from 27.1 to 126.5 mg g-1."

REF [53] was added:

[53] H Javadian, M. Ruiz, M. Taghvai, A.M. Saestre. Novel magnetic nanocomposite of calcium alginate carrying poly(pyrimidine-thiophene-amide) as a novel green synthesized polyamide for adsorption study of neodymium, terbium, and dysprosium rare-earth ions. Colloids and Surfaces A: Physicochemical and Engineering Aspects 603 (2020) 125252. DOI: 10.1016/j.colsurfa.2020.125252.

23) Please provide the activation energy for Nd biosorption.

Dear reviewer, the activation energy is just important to establish the time to attain the equilibrium. This part was already evaluated by t0.5 and t0.95. However, some authors compare Ea with DG° or DH°, this information is completely wrong in the papers. The activation energy is kinetic, and G° or DH° are thermodynamic parameters. Therefore, it should not be compared.

24) Please check the Kg and Keº values for LEB-18 strain as well as the Kg, Keº and DGº values for LEB-52 strain. They are the same and this is wrong.

There is nothing wrong in the paper. All calculations were based on the references:

[19] E.C. Lima, M.H. Dehghani, A. Guleria, F. Sher, R.R. Karri, G.L. Dotto, H.N. Tran, CHAPTER 3 - Adsorption: Fundamental aspects and applications of adsorption for effluent treatment, in: Hadi Dehghani, M., Karri, R., Lima, E. (Eds.), Green Technologies for the Defluoridation of Water. Elsevier, 2021, 41–88. Doi:10.1016/B978-0-323-85768-0.00004-X.

[38] E.C. Lima, A. Hosseini-Bandegharaei, J.C. Moreno-Piraján, I. Anastopoulos. A critical review of the estimation of the thermodynamic parameters on adsorption equilibria. Wrong use of equilibrium constant in the Van't Hoof equation for calculation of thermodynamic parameters of adsorption. J Mol Liq 273 (2019) 425-434. Doi: 10.1016/j.molliq.2018.10.048.

In the supplementary material is given:

"Thermodynamic studies for the Nd(III) adsorption onto LEB-18 and LEB-52 Spirulina platensis algae adsorbents were performed at temperatures ranging from 22ºC to 55°C (298 to 328 K).

The Gibb's free energy change (∆G0, kJ mol-1), enthalpy change (∆H0, kJ mol-1), and entropy change (∆S0, J mol-1K-1) were evaluated with the aid of Equations 14-17, respectively [17,36].

(14)

(15)

(16)

The combination of Equations 14 and 15 leads to equation 17

(17)

            R is the universal gas constant (8.314 J K-1 mol-1); T is the absolute temperature (Kelvin); Mw is the molecular weight of the adsorbate (g mol-1),  is the standard molar concentration of the adsorbate, which by definition is 1 mol L-1; g is the activity coefficient of the adsorbate. is the thermodynamic equilibrium constant, calculated according to equation 16.  is dimensionless [17,36].

            is calculated by converting Kg values (Liu equilibrium constant) or KL (Langmuir equilibrium constant), expressed in L mg-1 into L mol-1. Firstly, the value Kg or KL  is multiplied by 1000 (mg g-1), and then multiplied by the molecular weight of the adsorbate (g mol-1) and by the standard concentration of the adsorbate (1 mol L-1) and divided by the activity coefficient of the adsorbate (g- dimensionless) [17,36]. It is assumed that the solution is sufficiently diluted to consider that the g is unitary [17,36]. Making these calculations,  becomes dimensionless [17,36].

Equation 17 is the linearized van't Hoff equation [17]. On the other hand, Lima et al. [17] recently proposed using the nonlinear van't Hoff equation, as presented in equation 18."

(18)

25) To recover Nd from Nd-loaded biosorbents, desorption studies are required. Please provide desorption data.

We understand the viewpoint. However, the objective here is to concentrate Nd. This way, after the biosorption, the biomass loaded with Nd needs to be burned, then concentrating Nd. No desorption is required.

Reviewer 3 Report

Abstract need to be rewritten, in the present form it is too general. It should contain short introduction, material and methods and main results.

The difference between applied spirulina strains need to be evidenced. Authors should explain why they used these strains.

In paragraph 2.1 authors wrote that zero point charge (pHzpc) of S.platensis was determined. Please add these data to the results.

SEM images are not very informative in describing differences between strains.

Add FTIR spectra of biomass after sorption

Add units in table 1.

There is an error on Fig 4. The name of PFO written is indicated twice on each picture.

There is no discussion of the results in the paper.

Authors need to add information related to the comparison of spirulina with other sorbents used for Nd removal.

Author Response

26) The abstract needs to be rewritten; in the present form, it is too general. It should contain a short introduction, material, methods, and main results.

Dear reviewer, please read the abstract, it presents a short introduction:" Rare earth elements such as Neodymium (Nd) are important elements used mainly in developing new technologies. Although they are found in low concentrations in nature, they can be obtained by extracting solid samples such as phosphogypsum"

Materials and methods: "In this work, two strains of Spirulina platensis (LEB 18 and LEB 52) were employed as biosorbents for efficiently removing the Nd element from the aqueous media."

Main results: "Biosorption tests were carried out in a batch system, and the results of the biosorption kinetics showed that for both materials, the biosorption of Nd was better described by the Avrami model. Besides, it could be considered that 80 min would be necessary to attain the equilibrium of Nd(III) using both biosorbents. The result of the biosorption isotherms showed that for both strains, the best-fitted model was the Liu model, having a maximum biosorption capacity of 72.5 mg g-1 for LEB 18 and 48.2 mg g-1 for LEB 52 at a temperature of 298 K. Thermodynamics of adsorption showed that for both LEB-18 and LEB-52 the process was favorable (∆G° < 0) and exothermic (∆H° -23.2 for LEB-18 and ∆H° -19.9 for LEB-52). Finally, both strains were suitable to uptake Nd, and the better result of LEB 18 could be attributed to the high amount of P and S groups in this biomass. Based on the results, a mechanism of electrostatic attraction of Nd3+ and phosphate and sulfate groups of both strains of Spirulina platensis was proposed."

 27) The difference between applied Spirulina strains needs to be evidenced. The authors should explain why they used these strains.

This difference is clearly presented in Table 2

Table 2. Elemental composition of LEB-18 S. platensis and LEB-52 S. platensis biomasses.

Element

LEB-18* (%)

LEB-52* (%)

C

44.2±2.1

45.7±1.5

H

11.2±0.3

10.7±0.2

N

24.3±1.0

31.3±1.5

O

9.1±0.5

9.3±0.2

P

6.7±0.1

1.7±0.1

S

4.5±0.1

1.3±0.1

Also in the abstract of the paper:

"Finally, both strains were suitable to uptake Nd, and the better result of LEB 18 could be attributed to the high amount of P and S groups in this biomass. Based on the results, a mechanism of electrostatic attraction of Nd3+ and phosphate and sulfate groups of both strains of Spirulina platensis was proposed.”"

In the conclusion section of the paper:

"In addition to the results found in elemental analysis, the greater efficiency of LEB-18 concerning LEB 52 may be due to a greater presence of negative phosphate and sulfate groups, which can interact with positive Neodymium, facilitating electrostatic attraction and consequently increasing the biosorption capacity of the material."

28) In paragraph 2.1 authors wrote that zero point charge (pHzpc) of S.platensis was determined. Please add these data to the results.

This mistake was corrected in the revised manuscript; please see query #6 of reviewer #1.

29) SEM images are not very informative in describing differences between strains.

SEM is not informative for describing the difference between most biomass-based materials. Please see query #15 of reviewer #2.

30) Add FTIR spectra of biomass after sorption

Please see query #16 of reviewer #2.

31) Add units in table 1.

This mistake was corrected in the revised manuscript. Please see query #27.

32) There is an error in Fig 4. The name of PFO written is indicated twice on each picture.

This mistake was corrected; please see query #20 of reviewer #2.

31) There is no discussion of the results in the paper.

Rebuttal: This paper is very well discussed. The biosorbents were characterized by using textural characterization (surface area, total pore volume, SEM), quantitative elemental analysis (which is possible to quantify sulfate and phosphate groups), FTIR analysis, pHpzc, besides studies of kinetics, equilibrium at four different temperatures as well as the correct obtaining of thermodynamic equilibrium parameters. Based on these results, it was possible to establish a mechanism of adsorption. All these results were soundly discussed in the current paper.

32) Authors need to add information related to the comparison of Spirulina with other sorbents used for Nd removal.

See in the revised manuscript: "The maximum biosorption capacities were 72.5 mg g-1 for LEB 18, and 48.2 mg g-1 for LEB 52 at a temperature of 298 K. These values are competitive with other materials used to uptake Nd from aqueous solutions. For example, Javadian et al. [53] compared around 20 adsorbents used to uptake Nd from aqueous matrices. They found capacity values from 27.1 to 126.5 mg g-1."

REF [53] was added:

[53] H Javadian, M. Ruiz, M. Taghvai, A.M. Saestre. Novel magnetic nanocomposite of calcium alginate carrying poly(pyrimidine-thiophene-amide) as a novel green synthesized polyamide for adsorption study of neodymium, terbium, and dysprosium rare-earth ions. Colloids and Surfaces A: Physicochemical and Engineering Aspects 603 (2020) 125252. DOI: 10.1016/j.colsurfa.2020.125252.

Round 2

Reviewer 1 Report

The authors have made important changes to improve the quality of the manuscript. So, I recommend accepting the manuscript in this journal.

Author Response

The authors have made important changes to improve the quality of the manuscript. So, I recommend accepting the manuscript in this journal.

RESPONSE Thank you very much for the constructive evaluation of our work

Reviewer 2 Report

The authors have corrected and improved the manuscript according to my suggestions. I advise to publish the manuscript.

Author Response

The authors have corrected and improved the manuscript according to my suggestions. I advise to publish the manuscript.

RESPONSE Thank you very much for the constructive evaluation of our work

Reviewer 3 Report

In the manuscript is not indicated from where spirulina strains were obtained.

Do I understand correctly that beside content of P and S there is no difference between two spirulina stains?

If SEM and FTIR according to authors do not provide any information about metal adsorption what was the reason to include these data in the manuscript?

How authors determined that namely phosphate and sulfate groups play main role in Nd sorption?

“This trend could be attributed to the differences between the surface groups of the biomasses. LEB-52 contains more N (Table 1), arranged in NH2 and easily converted to NH3+, repealing the Nd3+ positive ions. On the other hand, LEB-18 contains more P and S (Table 1), which forms PO4 3- and SO4 2- attracting the Nd3+ or complexing it.” — Please add references confirming this affirmation

Please add in the manuscript information at which pH value Nd starts to precipitate, since it is know what for other rare earth elements is pH 6

Author Response

Thank you for the constructive evaluation of our work. See the responses in detail:

1-In the manuscript is not indicated from where spirulina strains were obtained.

RESPONSE Please, this information is detailed in references 27 and 28.

2-Do I understand correctly that beside content of P and S there is no difference between two spirulina stains?

RESPONSE There are other differences, but the main difference that affects adsorption is the P and S content.

3- If SEM and FTIR according to authors do not provide any information about metal adsorption what was the reason to include these data in the manuscript?

RESPONSE It was inserted to present biosorbent features in better detail.

4- How authors determined that namely phosphate and sulfate groups play main role in Nd sorption?

RESPONSE: It was based on the literature [27, 28, 30], characterization techniques and experimental biosorption results

“This trend could be attributed to the differences between the surface groups of the biomasses. LEB-52 contains more N (Table 1), arranged in NH2 and easily converted to NH3+, repealing the Nd3+ positive ions. On the other hand, LEB-18 contains more P and S (Table 1), which forms PO4 3- and SO4 2- attracting the Nd3+ or complexing it.” — Please add references confirming this affirmation

RESPONSE: It was inserted [27, 28, 30]

Please add in the manuscript information at which pH value Nd starts to precipitate, since it is know what for other rare earth elements is pH 6

RESPONSE: In the conditions of this study, with these reagents, the precipitation starts at pH higher than 7. The information was added to the manuscript